# Particle Image Velocimetry of Oil–Water Two-Phase Flow with High Water Cut and Low Flow Velocity in a Horizontal Small-Diameter Pipe

**DOI:** 10.3390/s19122702

**Published:** 2019-06-16

**Authors:** Lianfu Han, Haixia Wang, Xingbin Liu, Ronghua Xie, Haiwei Mu, Changfeng Fu

**Affiliations:** 1College of Electronics Science, Northeast Petroleum University, Daqing 163318, China; lianfuhan@nepu.edu.cn (L.H.); Wanghaixia_1994@163.com (H.W.); muhaiwei_123@163.com (H.M.); 2Logging and Testing Services Company, Daqing Oilfield Limited Corporation, Daqing 163310, China; liuxingbin0459@163.com (X.L.); xieronghua0459@163.com (R.X.)

**Keywords:** particle image velocimetry, velocity, kriging interpolation, displacement sub-pixel fitting, horizontal well, oil–water two-phase flow

## Abstract

Velocity and flow field are both parameters to measure flow characteristics, which can help determine the logging location and response time of logging instruments. Particle image velocimetry (PIV) is an intuitive velocity measurement method. However, due to the limitations of image acquisition equipment and the flow pipe environment, the velocity of a horizontal small-diameter pipe with high water cut and low flow velocity based on PIV has measurement errors in excess of 20%. To solve this problem, this paper expands one-dimensional displacement sub-pixel fitting to two dimensions and improves the PIV algorithm by Kriging interpolation. The improved algorithm is used to correct the blank and error vectors. The simulation shows that the number of blank and error vectors is reduced, and the flow field curves are smooth and closer to the actual flow field. The experiment shows that the improved algorithm has a maximum measurement error of 5.9%, which is much lower than that of PIV, and that it also has high stability and a repeatability of 3.14%. The improved algorithm can compensate for the local missing flow field and reduce the requirements related to the measurement equipment and environment. The findings of this study can be helpful for the interpretation of well logging data and the design of well logging instruments.

## 1. Introduction

Oil exploitation has already entered the stage [1] of high water cut and low liquid yield in many oil fields, such as Daqing Oilfield and Jilin in China. For example, there are 46,000 oil wells in the Daqing oilfield of the Changyuan old area, with an average of 20.1 m^3^/day per well and a water cut of 94.6%, prompting adjustments [2,3] in oilfield development plan and technology. Horizontal wells, with small wellbore and chemical flooding techniques [4,5,6], are often used to improve oil recovery. The flow pattern of oil–water two-phase flow presents the mixed state of a dispersed upper layer of oil in water above an oil-free water layer, resulting in oil–water slippage. The oil–water flow state is affected slightly by this slippage after implementing the new technology, creating difficulties in interpreting logging data [7,8] and assessing logging instruments in research development. Interpreting data and the accuracy of instruments are largely dependent on flow characteristics, which are parameterized by the flow pattern, flow field, velocity, and components. The measurements of velocity and the velocity vector field are indispensable parts of the dynamic monitoring of an oilfield development, which provides an important reference for flow measurements in oilfield exploitation and improving oil recovery. Therefore, studying oil–water two-phase flow with high water cut and low flow velocity, particularly in a horizontal small-diameter pipe, is necessary and timely. Velocity measurements of oil–water two-phase flow may be performed by various methods, including the ultrasonic method [9,10,11], the conductance method [12,13], the capacitance method [14,15,16], the electromagnetic method [17], the thermal tracer method, the optical fiber method, the capacitance resistance tomography method [18], the ultraviolet fluorescence [19,20], and particle image velocimetry (PIV) [21,22,23,24]. PIV has become an important research method because of its field visualization capability [25,26]. PIV adopts an online approach with non-contact velocity measurements to realize real-time monitoring and non-interference measurements of the entire flow field [27,28]. It is used not only to analyze quantitatively the velocity vector field of the entire pipeline flow, but also to remove limitations associated with traditional single-point measurements. In 1996, Zachos et al. [29] proposed the application of PIV technology to multiphase flow research. Subsequently, Wang et al. [30] also studied PIV technology for two-phase flow. Wan et al. [31] analyzed the application of PIV technology in the field of gas–liquid, gas–solid, and liquid–solid two-phase flow. Huang et al. [32] first proposed image deformation for PIV technology. Scarano et al. [33] proposed an iterative image deformation algorithm to improve the accuracy of particle matching. Wang et al. [34] studied the fast Fourier transform (FFT)-based cross-correlation algorithm and analyzed its characteristics. Li et al. [35] introduced seven speed extraction methods for digital PIV (DPIV) technology and analyzed their characteristics and limitations. Xu et al. [36] improved the PIV cross-correlation algorithm based on a thorough analysis of the conventional PIV algorithm. Xu et al. [37] developed a PIV experimental system for oil–water two-phase flow, and explained that in the case of horizontal dispersion flow, small oil droplets have good traceability, indicating that they can replace tracer particles for PIV measurement. Gollin et al. [38] reviewed the performance of PIV technology and particle tracking velocimetry (PTV) technology in measuring particle flow characteristics. Scharnowski et al. [39] used PIV and PTV technologies to analyze the free-stream flow at the Trisonic Wind tunnel Munich, and realize the accurate measurement of turbulence. Cerqueira et al. [40] improved the PTV algorithm to overcome the phenomenon of bubble overlap in backward lighting, and used this method to analyze upper laminar flow and turbulent bubbles.

There are two main factors affecting PIV measurement accuracy: (1) measuring equipment, and (2) the PIV algorithm. Current research focuses on improving the performance of hardware, such as using faster and higher-resolution cameras, making glass masks to prevent cylindrical optical distortion, and making particle tracers without trailing tails. In the PIV image processing algorithm, a few scholars (Laloš et al. [41]) applied the spline interpolation method to the PIV algorithm to analyze the measured values and determine the locus of points in the transient process. Foucaut et al. [42] used Whitaker interpolation and other algorithms to obtain the best iteration results), and have also noticed the impact of interpolation on the accuracy of the algorithm. However, the linear interpolation algorithm is generally adopted, and the independent *X* and *Y* directions are mostly adopted for displacement sub-pixel fitting of the two-dimensional flow field, so the measurement accuracy is not high. In this paper, Kriging interpolation and two-dimensional displacement sub-pixel fitting are used to improve the measurement accuracy and reduce the performance requirements of the measuring equipment.

A series of studies on the influence of optical distortion correction and linear interpolation on the PIV algorithm have promoted the development of PIV technology, but they have high requirements for measuring equipment and the flow environment. In using PIV to study the flow characteristics of the fluid with high water cut and low flow velocity in a horizontal small-diameter pipe, the limitations of image acquisition equipment and flow environment make the analysis of velocity vectors prone to producing blank and error vectors; that is, velocity vectors are unassigned or with large errors in the direction and magnitude of vectors, as shown in Figure 1. This is especially obvious in the image processing of oil–water two-phase flow with low flow velocity in a horizontal small-diameter pipe. To solve this problem, this paper expands the one-dimensional displacement sub-pixel fitting to two dimensions, and improves the PIV algorithm by Kriging interpolation [43,44,45]. This not only improves the accuracy of velocity vector field measurements of oil–water two-phase flow, but also provides a foundation for the interpretation of logging data and evaluating logging instruments for further research and development. This method can be used to estimate the time and distance of flow field stabilization in a horizontal pipe to determine the measuring position and response time of the logging tool.

The outline of this paper is as follows. First, the principle of the PIV algorithm is presented. Second, the PIV algorithm is improved by establishing the two-dimensional displacement sub-pixel fitting model and incorporating the Kriging interpolation in the velocity vector algorithm. Then, the improved PIV algorithm is verified through simulation and experiments. Finally, with the improved PIV algorithm, the velocity vector field is obtained for oil–water two-phase flow with a high water cut and low flow velocity in a horizontal small-diameter pipe.

## 2. PIV Algorithm Improvement

### 2.1. Principle of PIV Algorithm 

As shown in Figure 2, the PIV velocity measurement system consists of five parts: tracer particles (the small oil droplets), the measured flow field, a light source, a high-speed camera, and a computer for data acquisition. The high-speed camera takes images continuously of the oil–water two-phase flow. By analyzing and processing the captured images, the vector displacement of the tracer particles between two successive frames is obtained. The velocities |*v*| of tracer particles are calculated to give the fluid velocities at each point, as shown in Equation (1):(1)$$\left|v\right|=\underset{\Delta t\to 0}{\lim}\frac{\sqrt{{\left(\Delta x\right)}^{2}+{\left(\Delta y\right)}^{2}}}{\Delta t}$$ where Δ*x* and Δ*y* are the displacement of tracer particles in the *X* and *Y* directions, respectively, and Δ*t* is the time difference between the two frames of the images.

The overall process of PIV algorithm is shown in Figure 3, and two consecutive frames of images are preprocessed [47]. This involves: (1) image cropping to select useful areas of information and remove areas of irrelevant information; (2) grayscale conversion; and (3) contrast enhancement. The Iterative Closest Point (ICP) technique [48,49] is used to achieve precise pairing of particles, so that the size of droplet hardly affects the measurement accuracy. Second, a preliminary velocity vector image (the velocity vector image obtained by the PIV algorithm has a large number of blank and error vectors, which are not processed and corrected) is obtained by the processing of the grayscale images, which includes calculating displacement cross-correlations, detecting peaks in cross-correlations, performing a displacement sub-pixel fitting, and resolving the velocities. Last, from the preliminary velocity vector image, an error vector is filtered out using the signal-to-noise ratio (SNR), the peak, global, and local filters, and linear interpolation to produce a corrected velocity vector image.

### 2.2. PIV Algorithm Improvement Method

To correct the error vectors of velocity vector field and improve the accuracy of the velocity measurement, the PIV algorithm is improved in two aspects: establishing a two-dimensional displacement sub-pixel fitting model and developing an interpolation algorithm for the velocity vectors.

#### 2.2.1. Establishment of Two-Dimensional Displacement Sub-Pixel Fitting Model

During PIV analysis, the image information is converted into digital information, and a cross-correlation algorithm is used to calculate the displacement vectors of the tracer particles. Let the functions of the grayscale image of the *n*^th^ frame and (*n*+1)^th^ frame be denoted *f*(*x*, *y*) and *g*(*x*, *y*), respectively. The image functions of the two frames are then cross-correlated. To simplify the calculation and reduce the influence of white noise—the power spectrum density of which is assumed to be distributed uniformly over the whole frequency domain of the measurement results—the "similarity" measurement of general template matching is normalized to the standardize cross-correlation function [50], and the cross-correlation function *ϕ*(*m*, *n*) is shown as Equation (2):(2)$$\varphi (m,n)=\frac{\sum_{k=-\infty }^{\infty }\sum_{l=-\infty }^{\infty }f(k,l)g(k+m,l+n)}{\sqrt{\sum_{k=-\infty }^{\infty }\sum_{l=-\infty }^{\infty }f^{2}(k,l)\sum_{k=-\infty }^{\infty }\sum_{l=-\infty }^{\infty }g^{2}(k,l)}}$$
where *k* and *l* are the respective horizontal and vertical indices of a pixel in the interrogation window, and *m* and *n* are the respective index differences for the horizontal and vertical indices of the two chosen pixels one from each frame image.

From the property of the cross-correlation function *ϕ*(*m*, *n*), the larger its value, the more relevant the interrogation windows of the two image frames. Therefore, the whole pixel size of the displacement of pixels in the interrogation window [51] is shown as Equation (3):(3)X={(m,n)|maxφ(m,n)}

As shown in Figure 4, because the gray level function is a discrete sequence, the peak-to-peak correlation value *ϕ*(*m*, *n*) and the displacements *m* and *n* corresponding to the peak values obtained from the cross-correlation process are both integer multiples of pixel indies. The integer multiples of pixels are not the best match with the actual displacement in the two frames of images in the actual process. Therefore, a displacement sub-pixel fitting is required. At present, displacement sub-pixel fitting involves fitting *X* and *Y* directions independently, and then combining them. If displacement sub-pixel fitting is performed in *X* and *Y* directions respectively, the displacement sub-pixel fitting obtained in each direction is the best. However, if the displacement sub-pixel fitting in *X* and *Y* directions is directly combined, it may not be the best displacement sub-pixel fitting for this point. This method is simple and practical, but the flow field is not correlated in all directions. So, the fitting precision is not high. In this paper, the displacement sub-pixel fitting is directly carried out for the whole two-dimensional graph, and the displacement sub-pixel fitting of a certain point is obtained, which is the best displacement sub-pixel fitting for this point. For this reason, a two-dimensional pixel fitting model based on a Gaussian image distribution is established.

As the distribution of the image formed by the light source beam is assumed to be Gaussian [51,52], the correlation coefficient in the surrounding area at point *Q*_0_(*x*_0_, *y*_0_) is *R*_0_(*x*_0_, *y*_0_), and its two-dimensional distribution function [53,54] can be expressed as Equation (4):(4)$$R(x,y)=Ae^{\left(-\frac{{\left(x-x_{0}\right)}^{2}}{2\sigma_{x}^{2}}-\frac{{\left(y-y_{0}\right)}^{2}}{2\sigma_{y}^{2}}\right)}$$
where *A* is the correlation peak value, and *σ_x_*^2^ and *σ_y_*^2^ are the variances of the correlation peaks in the *X* and *Y* directions, respectively.

The maximum correlation coefficient around the record point *Q*_0_(*x*_0_, *y*_0_) is *R*(*x_m_*, *y_m_*), and its corresponding point is point *Q_m_*(*x_m_*, *y_m_*); *x_m_* and *y_m_* are the coordinate values of the displacement integer pixel in the *X* and *Y* directions. *Q_m_*(*x_m_*, *y_m_*) and its four adjacent points *Q_m_*_-1*m*_(*x_m_* − 1, *y_m_*), *Q_m_*_+1*m*_(*x_m_* + 1, *y_m_*), *Q_mm_*_−1_(*x_m_*, *y_m_* − 1), and *Q_mm_*_+1_(*x_m_*, *y_m_* + 1) are substituted into Equation (4), and a logarithmic transformation is performed to eliminate parameters *A*, *σ_x_*^2^, and *σ_y_*^2^ (see Appendix A) to obtain the exact location of the relevant peak-to-peak values, as shown in Equation (5):(5)$$\left\{ \begin{array}{l} x_{0}=x_{m}+\frac{\ln R(x_{m}-1,y_{m})-\ln R(x_{m}+1,y_{m})}{2(\ln R(x_{m}-1,y_{m})+\ln R(x_{m}+1,y_{m})-2\ln R(x_{m},y_{m}))}\\ y_{0}=y_{m}+\frac{\ln R(x_{m},y_{m}-1)-\ln R(x_{m},y_{m}+1)}{2(\ln R(x_{m},y_{m}-1)+\ln R(x_{m},y_{m}+1)-2\ln R(x_{m},y_{m}))} \end{array}\right.$$

#### 2.2.2. Improvement of Velocity Vector Interpolation Algorithm

The preliminary velocity vector image is obtained after the displacement cross-correlation processing and displacement sub-pixel fitting processing. Since the acquisition equipment and the environment within the flow pipe have limitations, the acquired image has a reduced quality. The preliminary velocity vectors are affected by noise, so they are prone to blank and error vectors in PIV processing.

To correct these error vectors and fill blank vectors, a linear interpolation algorithm is often used to replace the error vectors and interpolate blank vectors to improve the accuracy of each velocity measurement. As shown in Figure 5, the linear interpolation is performed using the velocity vector values of adjacent pixel points. The flow field function near blank vector Point *A* only has a continuous first derivative in Figure 5a. According to the velocity values of points *Q*_1_ and *Q*_2_, the interpolation error of linear interpolation is low, and the processing speed is fast. Hence, the linear interpolation method is feasible. The flow field function near the blank vector Point *A* has multiple continuous derivatives in Figure 5b. With the velocity values of Points *Q*_1_ and *Q*_2_, the linear interpolation leads to a large error *R_T_*. The error *R_T_* means that the corrected velocity vector image may not be smooth, and thereby affecting the accuracy of velocity measurements and exceeding their allowable error range. Hence, the linear interpolation method is not feasible.

The flow field of oil–water two-phase flow with high water cut and low flow velocity in a horizontal small-diameter pipe is affected by various accidental factors, and the function for the blank vector has multiple continuous derivatives. Therefore, using the linear interpolation to improve the measurement accuracy doesn’t work very well. To improve measurement accuracy, the Kriging interpolation is introduced into PIV velocity vector interpolation to improve the PIV algorithm. Given the known pixel velocity values in the interpolation region, the structural characteristics of the spatial position distribution of the pixel points in the blank vector and the known pixel points—as well as the structural information of the variogram—were observed, and an optimal unbiased linear estimation of the pixel velocity values in the blank vector was achieved. The Kriging interpolation algorithm [55] is characterized so that it takes into account not only the distance relationship between interpolated pixel points and other known pixel points, but also the autocorrelation relationship between known pixel points. The steps of the improved algorithm are as follows:

In the Kriging interpolation algorithm, the key to interpolating blank vectors is to find the weight value *λ* corresponding to each known pixel point. Using the unbiased optimal characteristics of the kriging interpolation, the Lagrangian algorithm [55] is adopted to obtain Equation (6):(6)$$F=2\sum_{i=0}^{n}\lambda_{i}\gamma_{ic}-\gamma_{cc}-\sum_{i=0}^{n}\sum_{j=0}^{n}\lambda_{i}\lambda_{j}\gamma_{ij}-2\mu (\sum_{i=0}^{n}\lambda_{i}-1)$$ where *γ_ij_* is the variation function between points *Q_i_*(*x_i_*, *y_i_*) and *Q_j_*(*x_j_*, *y_j_*), and −2*μ* is the Lagrangian multiplier. The partial derivative [56] of Equation (6) is calculated to obtain the solution of *λ*:(7)$$\left[\begin{array}{ccccc} \gamma_{11} & \gamma_{12} & \cdots & \gamma_{1n} & 1\\ \gamma_{21} & \gamma_{22} & \cdots & \gamma_{2n} & 1\\ \;\vdots & \vdots & \ddots & \vdots & \vdots \\ \gamma_{n1} & \gamma_{n2} & \cdots & \gamma_{nn} & 1\\ 1 & 1 & \cdots & 1 & 0 \end{array}\right]\left[\begin{array}{c} \lambda_{1}\\ \lambda_{2}\\ \;\vdots \\ \lambda_{n}\\ \;\mu \end{array}\right]=\left[\begin{array}{c} \gamma_{1c}\\ \gamma_{2c}\\ \;\vdots \\ \gamma_{nc}\\ \;1 \end{array}\right]$$ where the variation function *γ_ij_* is related to the distance *h_ij_* between pixel points. With the known pixel points *Q*_1_(*x*_1_, *y*_1_), *Q*_2_(*x*_2_, *y*_2_), ... , *Q_n_*(*x_n_*, *y_n_*) in the velocity vector interpolation region, Equation (8) is used to calculate the distance *h_ij_* between the *i*^th^ and *j*^th^ pixel points:(8)$$h_{ij}=\sqrt{{\left(x_{i}-x_{j}\right)}^{2}+{\left(y_{i}-y_{j}\right)}^{2}}$$

Since the image distribution is Gaussian, the relation *γ*(*h*)−*h* between variation function *γ*(*h*) and distance *h* is determined from the fitting of the Gaussian model, as shown in Equation (9):(9)$$\gamma_{ij}=\left\{ \begin{array}{ll} 0 & h=0\\ C_{0}+C(1-e^{-\frac{{h_{ij}}^{2}}{a^{2}}}) & h>0 \end{array}\right.$$

With the relational expression *γ*(*h*)−*h*, all the variation functions *γ_ij_* in Equation (9) can be calculated, and the weight value *λ* corresponding to each pixel point is obtained as shown in Equation (10):(10)$$\left[\begin{array}{c} \lambda_{1}\\ \lambda_{2}\\ \vdots \\ \lambda_{n}\\ \mu \end{array}\right]=\left[\begin{array}{c} \gamma_{1c}\\ \gamma_{2c}\\ \vdots \\ \gamma_{nc}\\ 1 \end{array}\right]{\left[\begin{array}{ccccc} 0 & \gamma_{12} & \cdots & \gamma_{1n} & 1\\ \gamma_{21} & 0 & \cdots & \gamma_{2n} & 1\\ \vdots & \vdots & \ddots & \vdots & \vdots \\ \gamma_{n1} & \gamma_{n2} & \cdots & 0 & 1\\ 1 & 1 & \cdots & 1 & 0 \end{array}\right]}^{-1}$$

Therefore, the velocity vector value at the blank vector can be expressed as:(11)$${\hat{Z}}_{c}=\sum_{i=0}^{n}\lambda_{i}Z_{i}$$ where *Z_i_* is the velocity vector at the given pixel point.

To summarize, a two-dimensional displacement sub-pixel fitting model and a velocity vector interpolation algorithm based on Kriging were developed and applied to PIV algorithm to improve the accuracy of velocity measurement for this study. For convenience of discussion, the improved model is abbreviated as the 2D-KPIV model.

### 3.2. D-KPIV Model Validation

The 2D-KPIV model was experimentally verified by simulated particle images with complex flow fields. The 2D-KPIV and PIV models were used to calculate the velocity vector of each particle, and the absolute errors and relative errors of the two models were obtained by comparing them with velocity vectors obtained from theory.

Figure 6 is the simulated particle image. The green particles represent the original particles, and the red particles represent the particles obtained following the original particles under swirl flow, shear flow, and increased noise. The image on the right is an enlarged version of the black box on the left. The displacement vector of each point particle is known, and the time interval between two successive images is 0.01 s. The theoretical velocity vector image is obtained by using the displacement vector of each point and the time between images, as shown in Figure 7a.

Figure 7 gives the velocity vector images obtained by processing Figure 6, with color indicating the magnitude of the particle velocity increasing as the color changes from blue to red; the arrows gives the direction of particle movement. Figure 7 shows the simulated velocity vector image obtained from theory, using the PIV algorithm, and applying the 2D-KPIV algorithm.

Regarding the velocity vector interpolation algorithm, the difference between the 2D-KPIV algorithm and PIV algorithm is that the former takes the pixel values of adjacent points and neighboring points as well as the distances between all the points, whereas the latter only takes the pixel values of adjacent points into account. The particle velocity encountered in swirling motion increases first and then decreases with an increasing rotation radius corresponding to the position of the particle in the swirl. With influences from other swirls, the velocities at the center of the vortex are greater than those between the two sides, which conforms to the characteristics of the particle swirling motion. As evident in Figure 7, the velocity vector image processed by the 2D-KPIV algorithm is smoother on the whole than that processed by PIV algorithm, and closer to the simulated velocity vector image obtained from theory.

A comparison of the absolute and relative errors in the calculation results of the two models with those of the actual results is shown in Table 1. The relative error δ is calculated by Equation (12):(12)$$\delta =\frac{cv-tv}{tv}\times 100\%$$ where *cv* is the calculated value and *tv* is the real value. Therefore, the measurement accuracy of the 2D-KPIV algorithm is higher than that of PIV algorithm, indicating that the 2D-KPIV algorithm is accurate and more feasible.

## 4. Experiment

To verify the reliability and measurement accuracy of the 2D-KPIV algorithm, the experiment platform as shown in Figure 8 was designed. It mainly includes: (1) a high-speed camera; (2) an oil and water circulation system comprising separate oil and water storage tanks, mixing devices and separation tanks; (3) a steady flow section and test section; and (4) a flow control module for the oil and water that includes a computer, controller, actuator, and measuring system. According to measurement requirements, the control system sends different proportions of oil and water to the mixed flow section. After achieving a steady flow in the steady flow section, oil–water two-phase flow in a horizontal small-diameter pipe is basically in a stable state. The camera captures the images of the fluid under the illumination of a light source, and the images are processed using PIV and the 2D-KPIV algorithm.

The experiment platform as shown in Figure 9 was built. The following specifications are as follows: (1) arrangement: horizontal; (2) pipe inner diameter: 20 mm; (3) pipe length of the steady flow part: 2 m; (4) pipe length of the test section: 0.1 m; (5) fluid: a mixture of tap water and white oil, white oil density: 0.8 g/cm^3^; (6) fluid conditions: velocity range: 0.184 to 0.737 m/s; water cut range: 85% to 99%; (7) camera: HSVISION Macrovis EoSens, frame speed: 0.001 s, image resolution: 1280 pixels × 1066 pixels.

The test section is a transparent glass tube, for which optical dispersion inevitably causes errors. Therefore, Snell’s law and the triangle theorem were used to correct these errors. Only the upper part of the pipe is used, because the pipe is axial-symmetric. The light path propagation diagram of a point on the PIV measurement plane is shown in Figure 10. The particle at Point *A*_0_’ in the captured image is actually a particle at Point *A*_0_. The correction is given by:(13)$$\left[X_{A_{0}}^{\prime }\;Y_{A_{0}}^{\prime }\right]=\left[X_{A_{0}}\;Y_{A_{0}}\right]\left[\begin{array}{cc} 1 & 0\\ 0 & \frac{n_{1}}{n_{3}\cos(\beta_{1}+\beta_{2}-\alpha_{1}-\alpha_{2})} \end{array}\right]$$
where *n*_1_ and *n*_3_ are the refractive indexes of the liquid in the test section and air respectively; *α_i_* and *β_i_* (*I* = 1, 2) are the angles of incidence and refraction of the light passing through different media, respectively. They obey the geometric Formula (13):(14)$$\left\{ \begin{array}{l} \alpha_{1}=\arcsin(\frac{Y_{A_{0}}}{R_{1}})\\ \alpha_{2}=\arcsin(\frac{n_{1}Y_{A_{0}}}{n_{2}R_{2}})\\ \beta_{1}=\arcsin(\frac{n_{1}Y_{A_{0}}}{n_{2}R_{1}})\\ \beta_{2}=\arcsin(\frac{n_{1}Y_{A_{0}}}{n_{3}R_{2}}) \end{array}\right.$$
where *n*_2_ is the refractive index of the glass, *R*_1_ and *R*_2_ are the inner and outer diameters of the pipe, respectively, and *Y_A_*_0_ is the ordinate value of the particle at point *A*_0_.

## 5. Results and Discussion

Figure 11 shows two consecutive frames of images randomly selected. The flow body to be measured is an oil–water two-phase flow with a water cut of 90%, and the average velocity is 0.553 m/s. The overall flow pattern corresponds to a state of oil in water. Due to the effect of gravity, the oil bubbles are mostly distributed above. However, oil bubbles are completely surrounded by water without stratification at this instant, because the water cut is high. The small oil bubbles have good flow properties, and therefore have good attributes as tracer particles.

Figure 12 shows the images in Figure 11 after preprocessing by extracting the useful information area, and applying gray-level conversion and contrast enhancement. Then, the 2D-KPIV and PIV algorithms were employed in the velocity vector field analysis and applied in the velocity measurement calculation of the two images in Figure 12. The results that are shown in Figure 13 are those for the preliminary velocity vector image, the velocity vector image using the PIV algorithm, and the velocity vector image using the 2D-KPIV algorithm. The images on the right include an enlarged version of the black box on the left. The color indicates the magnitude of the velocity, which increases as it changes from blue to red. The tail and the head of each arrow identify the start and end points with each movement.

Figure 13a shows that with a low flow velocity for the fluid to be measured, the flow pattern is in the mixed state of an upper layer of dispersed oil in water on a layer of oil-free water, resulting in an uneven distribution in oil bubbles in the horizontal pipe. In addition, the captured image itself contains noise and other factors, so that the preliminary velocity vector image obtained contains a large number of blank and error vectors. Figure 13b gives the velocity vector image obtained by applying the PIV algorithm to the image using SNR filtering, peak filtering, global filtering, local filtering, and linear interpolation. The average velocity of the fluid is 0.490 m/s, and its error is 10.9%. The error vectors are filtered using the filtering algorithm, and blank vectors are interpolated using the linear interpolation algorithm. Linear interpolation only considers the pixel values of adjacent points, so the corrected velocity vector image is not smooth, and the error is still large at more than 10%. Figure 13c shows the result of applying the 2D-KPIV algorithm; the error vectors are corrected, and the blank vectors are filled. The average velocity of the fluid is 0.521 m/s, and the error is 5.3%, which is better than that for PIV algorithm.

When the total fluid flow is very low, the oil phase is in an upwelling state, resulting in oil–water slippage. Having opted for a small pipe diameter and a steady-stage fluid flow stage, the influence of oil–water slippage is small, and does not affect the overall velocity measurement. Figure 13b,c rejects the obvious error vectors and interpolates blank vectors based on those in Figure 11a. The overall velocity vector field distribution shows that the velocities of the upper layer are greater than those of the lower layer, conforming the low flow velocity field distribution. Although there are no blank vectors in Figure 13b, the individual velocity vectors change abruptly both in direction and magnitude, so the velocity vector image in Figure 13b is not smooth. The 2D-KPIV algorithm interpolates blank vectors by taking into account the pixel values of the adjacent points and the pixel values of other adjacent points. The velocity vectors change gently in both direction and magnitude, making the overall velocity vectors in Figure 13c smooth and stable. This effectively removes the background noise of the image and reduces the measurement error of the velocity vectors. Moreover, the error is lower for the 2D-KPIV algorithm than for PIV algorithm, which proves that the algorithm is effective. Besides, the 2D-KPIV algorithm can compensate for the local missing flow field and reduce the requirement of PIV for measuring environment.

Table 2 shows the velocities and errors obtained by PIV algorithm based on linear interpolation and the 2D-KPIV algorithm. The true mean velocities are given by the control system in Figure 8. The maximum error of the latter is 5.8% and the average error is 2.68%, whereas the maximum error of the former is 10.9% and the average error is 6.27%. Both the maximum error and the average error have reduced. Moreover, a repeated experiment indicated a relatively stable trend with a high repeatability of 3.14%, which proves the reliability of the algorithm.

To mitigate the contingencies of the experiment, the total flow velocity and water cut were changed. The experiment results are shown in Figure 14. In Figure 14a,b, the maximum errors of each group using the 2D-KPIV algorithm were 5.6%, 5.9%, 5.9%, and 4.9%, and the average errors are 0.5%, 0.9%, 0.7%, and 1.1%, respectively. Whereas for PIV algorithm, the maximum errors are 19.5%, 34.1%, 26.3%, and 21.1%, and the average errors are 7.8%, 12.7%, 14.5%, and 13.9%, respectively. The errors for the 2D-KPIV algorithm are all within 6%, whereas the errors for the PIV algorithm are more than 10%. According to the above analysis, the velocity measured by the 2D-KPIV algorithm is not affected by flow velocity and water cut, and it is closer to the actual value. Therefore, the 2D-KPIV algorithm has high stability.

The PIV algorithm is changed in two regards: a two-dimensional displacement sub-pixel fitting and Kriging interpolation. This improves the measurement accuracy of the velocity. However, the improvement also increases complexity of the PIV algorithm and reduces the efficiency of the computation.

## 6. Conclusions

The effect of blank and error vectors on the measurement accuracy of a velocity vector field is studied for oil–water two-phase flow with high water cut and low flow velocity in a horizontal small-diameter pipe in this paper. To solve this problem, the PIV algorithm is improved in two aspects: a two-dimensional displacement sub-pixel fitting and Kriging interpolation; the improved PIV algorithm is named as the 2D-KPIV algorithm. The measuring accuracy of the 2D-KPIV algorithm is higher than 94.1%, and the reproducibility of the experimental data is 3.14%. This provides a feasible method for accurately drawing the oil–water two-phase flow field. The flow field can help estimate the time and the distance of flow field stabilization in a horizontal pipe, so the measuring position and response time of the logging tool is determined. The application of this work in a 3D-PIV technique of the well-log tomography should be further studied.

## Figures and Tables

**Figure 1 sensors-19-02702-f001:**
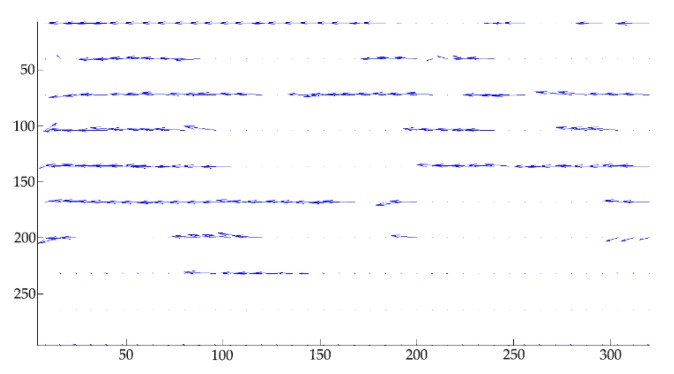
Velocity vector image with blank and error vectors.

**Figure 2 sensors-19-02702-f002:**
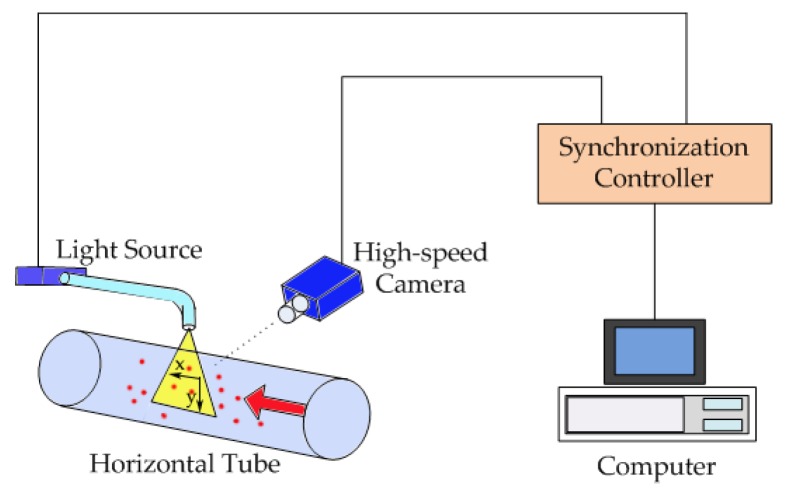
Schematic diagram of the particle image velocimetry (PIV) measurement system (adapted from Chen [46]).

**Figure 3 sensors-19-02702-f003:**
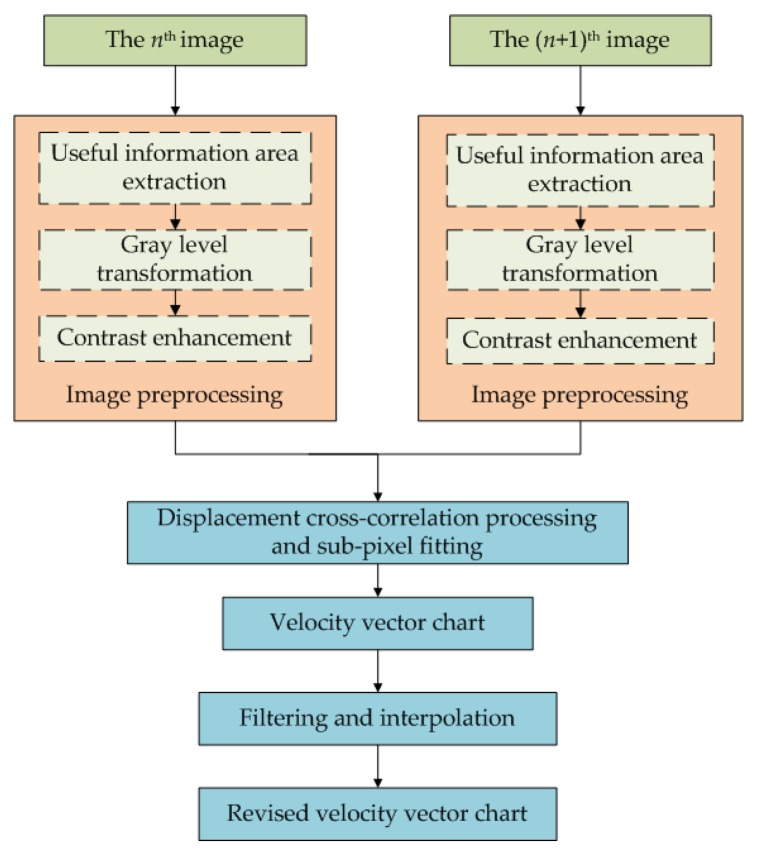
Particle image velocimetry (PIV) algorithm flow chart.

**Figure 4 sensors-19-02702-f004:**
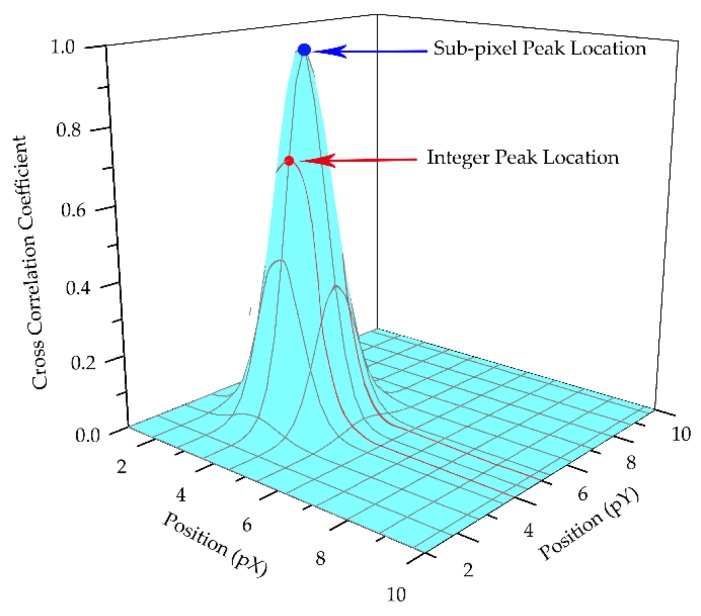
Displacement sub-pixel fitting schematic diagram.

**Figure 5 sensors-19-02702-f005:**
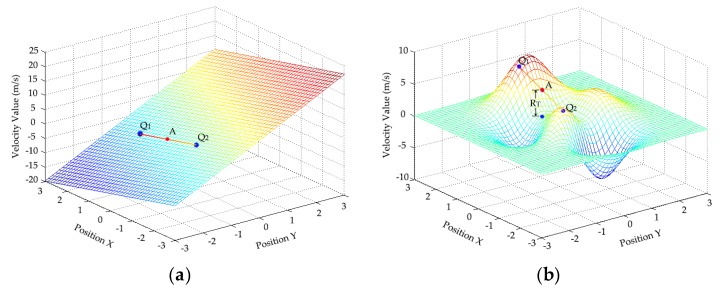
Errors caused by the interpolation of: (**a**) a single derivative; (**b**) multiple derivatives.

**Figure 6 sensors-19-02702-f006:**
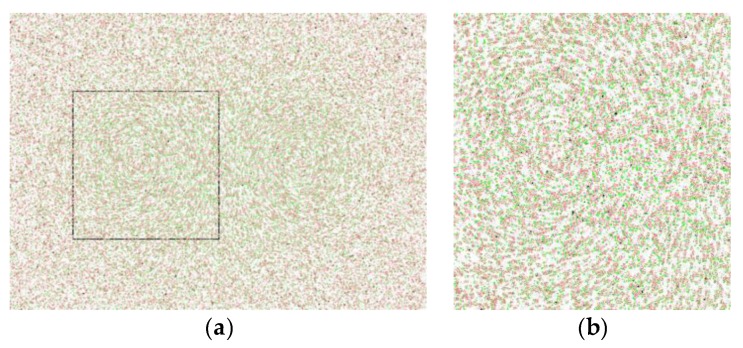
Simulated particle image: (**a**) Composite particle image; (**b**) locally enlarged image.

**Figure 7 sensors-19-02702-f007:**
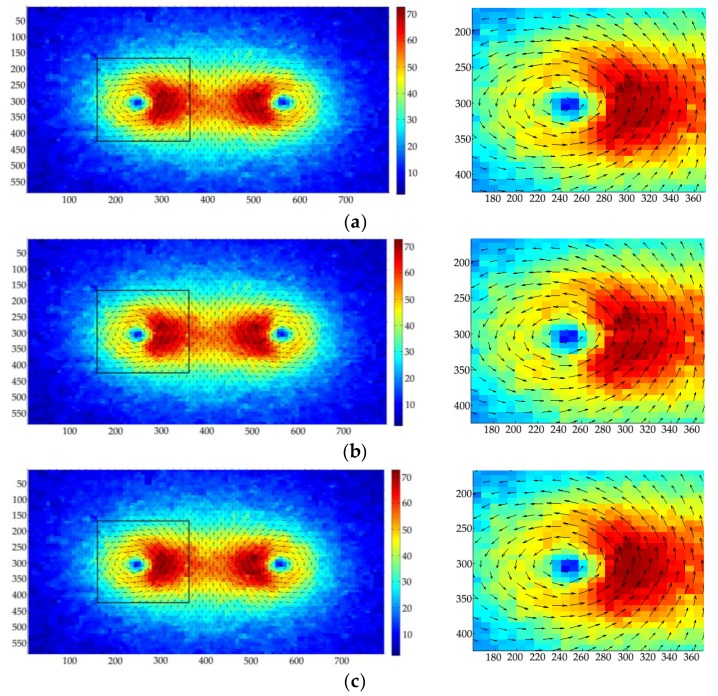
Velocity vector image: (**a**) from theory; (**b**) after using PIV algorithm; (**c**) after using the combined two-dimensional displacement sub-pixel fitting model and velocity vector interpolation algorithm based on Kriging (2D-KPIV).

**Figure 8 sensors-19-02702-f008:**
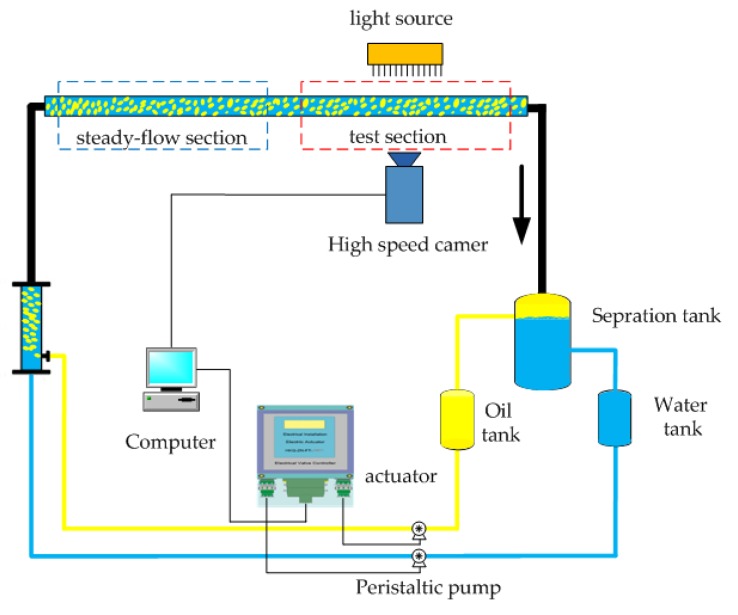
Schematic diagram of the experiment setup.

**Figure 9 sensors-19-02702-f009:**
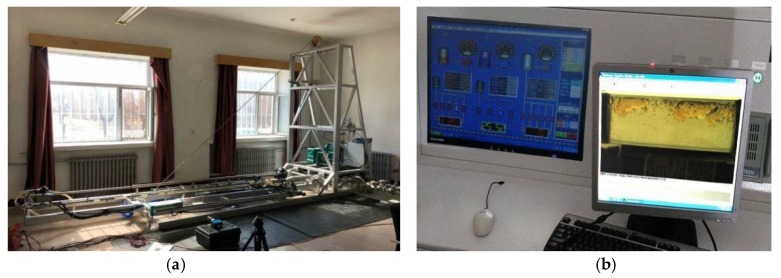
Experiment platform: (**a**) Image acquisition system; (**b**) control platform.

**Figure 10 sensors-19-02702-f010:**
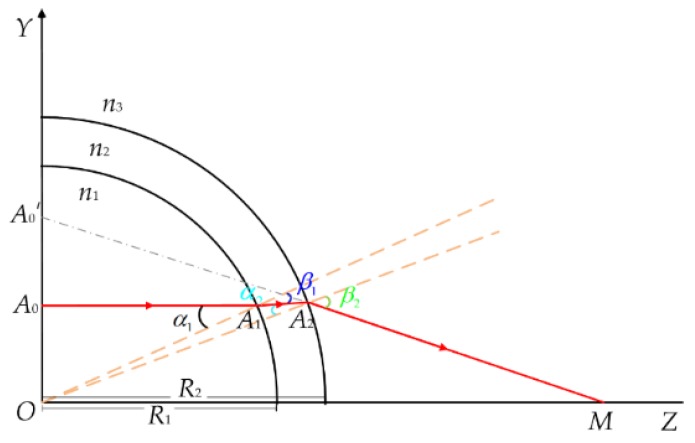
Schematic diagram of light path propagation.

**Figure 11 sensors-19-02702-f011:**
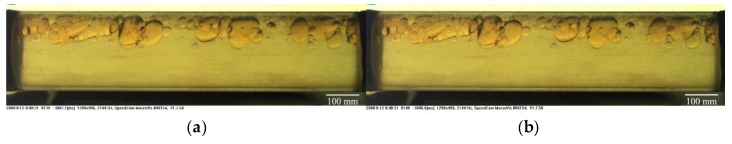
Raw image acquired from two consecutive frame images: (**a**) the *n*^th^ frame; and (**b**) the (*n+*1)^th^ frame.

**Figure 12 sensors-19-02702-f012:**
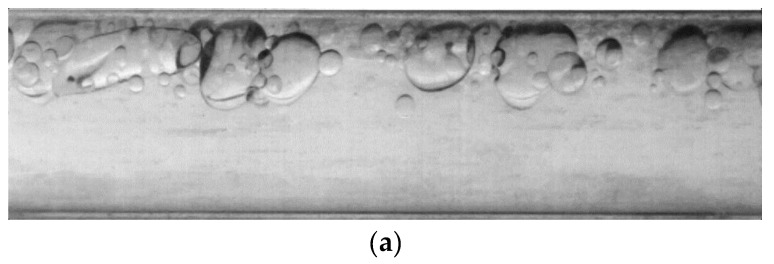
Two consecutive frames of preprocessed images: (**a**) the *n*^th^ frame; (**b**) the (*n+*1)^th^ frame.

**Figure 13 sensors-19-02702-f013:**
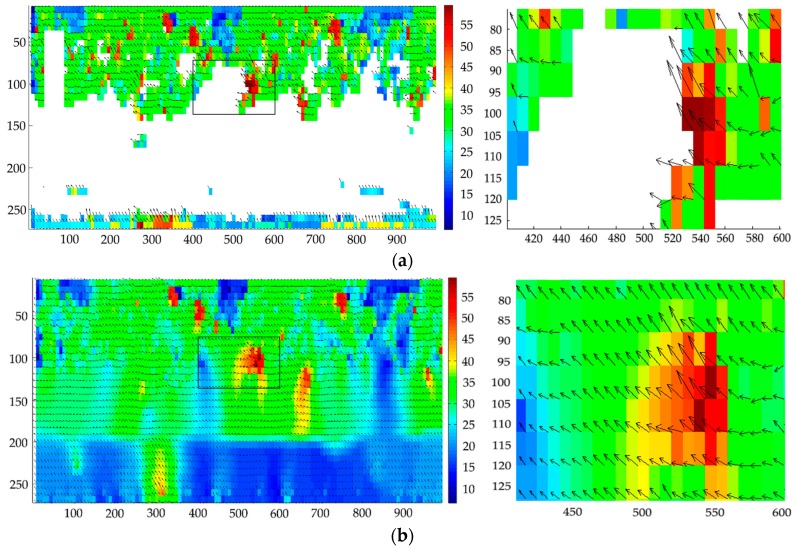
Velocity vector image: (**a**) preliminary; (**b**) after using the PIV algorithm; (**c**) after using the the 2D-KPIV algorithm.

**Figure 14 sensors-19-02702-f014:**
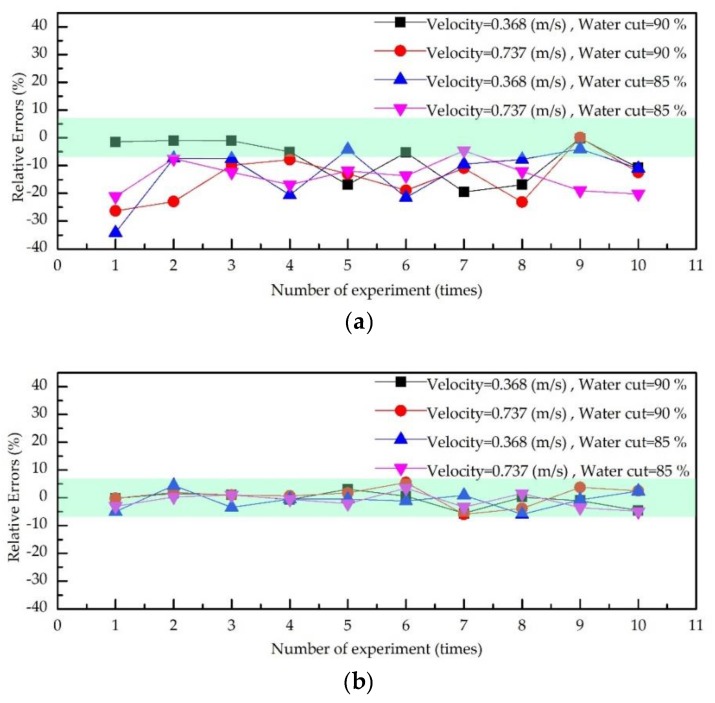
Errors diagrams of oil–water two-phase flow velocity measurement: (**a**) the PIV algorithm; (**b**) the 2D-KPIV algorithm

**Table 1 sensors-19-02702-t001:** Absolute errors and relative errors of particle image velocimetry (PIV) and the 2D-KPIV algorithm.

Model	Absolute Error (pixel/s)	Relative Error (%)
PIV	26.22	11.4
2D-KPIV	12.25	5.3

**Table 2 sensors-19-02702-t002:** Fluid velocities and measurement errors.

Serial Number	PIV Algorithm		2D-KPIV Algorithm	
	Velocity (m/s)	Errors (%)	Velocity (m/s)	Errors (%)
1	0.490	10.9	0.521	5.3
2	0.503	8.6	0.524	4.8
3	0.521	5.3	0.548	0.4
4	0.524	4.8	0.542	1.5
5	0.521	5.2	0.560	1.9
6	0.503	8.6	0.518	5.8
7	0.459	16.6	0.540	1.8
8	0.497	9.7	0.530	3.7
9	0.558	1.5	0.567	3.1
10	0.498	9.5	0.549	0.1
11	0.548	0.3	0.559	1.7
12	0.530	3.6	0.538	2.2
13	0.529	3.9	0.535	2.8
14	0.521	5.3	0.575	4.5
15	0.548	0.3	0.548	0.3

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
