# Peer review of "Particle Image Velocimetry of Oil–Water Two-Phase Flow with High Water Cut and Low Flow Velocity in a Horizontal Small-Diameter Pipe"

_sensors, 2019, doi:10.3390/s19122702_

Round 1

Reviewer 1 Report

I believe, the article needs to be improved (e.g. abstract, introduction, methods are discussed again in the results, in conclusion the authors should highlight the significance of this work in the wider context, including limitations and future studies). A separate file is attached to guide the authors to improve this manuscript.

Author Response

Response to Reviewer Comments

Manuscript number:  #sensors-5114461
Title:  "Velocity field measurement of oil–water two-phase flow with high water cut and low flow velocity in a small horizontal pipe based on PIV "
Correspondence Author: 
fuchangfeng8001@126.com

Dear Ms. April Zhu and Reviewer,

   Thank you for your letter and for the reviewers’ comments concerning our manuscript. Those comments are very helpful for revising and improving our paper.

We have made correction which we hope meet with approval. The response to reviewer’s comments is in the attachment. Besides, we use a professional English editing service to edit our manuscript.

Special thanks to you for your good comments.

We hope that these revisions are satisfactory and that the revised version will be acceptable for publication in the journal sensors. Thank you very much for all your help and looking

forward to hearing from you soon.
Wish you all the best!

Sincerely yours,
Changfeng Fu

Reviewer 2 Report

As attached.

Author Response

(The authors gave the same response as above.)

Reviewer 3 Report

Highlight changes in yellow in a next revision, please. No track changes.

Consider comments in the entire text.

I would ask the authors to revise the title in terms of being more concise and assertive and avoid abbreviations (also in keywords, if used alone…).

Abbreviations should be defined at first-use: abstract and again in the text

Please avoid duplicating information:

At the latter stage of oilfield development, the production fluid has the characteristics of 12 high water cut and low velocity

And then again:

At the latter stage of oilfield development, the production fluid has the characteristics of high 30 water cut and low flow velocity.

Check superscript use… “20.1 m3/d”

I do not see a single reference in the first extensive paragraph in the Introduction. Authors need to contextualize information…

1. Introduction 29

At the latter stage of oilfield development, the production fluid has the characteristics of high 30 water cut and low flow velocity. For example

At the start of a sentence relating an already defined abbreviation, the use of the references is not clear…exactly to what content does it relate?

PIV [14-16]

See that the use of direct reference style demands the need to insert the reference immediately after authors names (not at the end):

In 1996, A. Zachos, M. Kaiser et al. 52

Check all other cases…

Please do not start every sentence the same way…

… did this

. did that

Instead better contextualize the reference use to have a fluid text.

It starts at

In 1996, A. Zachos, M

(…)

And ends here:

“turbulent bubbles [28].”

Also see that authors names presentations differ…

Sven Scharnowski et al.

“Cerqueira, R.F.L.”

Revise all:

Last names and then reference number…

More duplicated content…

No added knowledge…

To solve these problems, this paper expands one-dimensional

To solve this problem, this paper expands the one-dimensional sub-pixel

Please check all inconsistencies in the entire text…

Section 2:

Assure all necessary citations are added immediately before known equations are presented

Then leave perfectly clear the original work and emphasize novelty…

All parameters must be defined after EACH equations and units added inside “()” where available…

Figure 1: do not define abbreviations again in captions:

“particle image velocimetry (PIV) system”

Assure fonts (size/type, etc) in figures are consistent all over… and smaller than the text

Reference information regarding figures must be added at the end of the caption, not in the text…

a high-speed camera and a computer [32-33].

See that already published figures add no knowledge and they must be removed if not adapted/modified and then state so, indicating authors names and reference number

Check all cases because then again:

Figure 2. Firstly, two consecutive frames of images are 93 preprocessed [34]

Again, write differently, add knowledge…:

The PIV algorithm flow chart

Figure 2. PIV algorithm flow chart

The last box is cut and again different font, compare to Figure 1…

Sections 2 and 3 should be merged…

2. The Principle of PIV

3. PIV model improvement

The relevance of content does not justify different sections

Perhaps adding section 4…

4. 2D-KPIV Model Validation

To present the impact of the text everywhere is not the way to do it, information must be at the end of introduction, last paragraph:

To correct the error vector of the velocity field and improve the accuracy of velocity 103 measurement, this paper improves the PIV algorithm for two aspects:

And English needs proofreading

Revise italics: “the X and Y directions,

Check if extensive mathematical information available in equations development should be presented as supplementary material

Clarify “the parameter is eliminated to

Captions must be self-explanatory. Better contextualize:

Figure 4. Error analysis chart: (a) The function has only one derivative; (b) The function has multiple derivatives. 174

Check all captions

[also in tables…

Table 1. Absolute Error and Relative Error”]

I would suggest revising the text to concentrate information because authors should “split” theses sentences through the text…

“To improve measurement accuracy, this paper introduces Kriging”

The expression “This paper” appears 9 times with duplicated information…

To improve measurement accuracy, this paper introduces Kriging

I must say that Figure 5 a) and b) look alike…

It is not expected hat readers have to use magnifying to see different things.

May be authors could emphasize areas? As done in Figure 6? Also, difficult but easier… to see

Table 1:

Present units inside “()”

“Absolute error/pixel/s Relative error/%”

[and compare to Table 2

Also in figures axis: check Figure 12…]

Section 5

A different heading is usually used, check samples:

5. Experiments

Avoid duplicating terms and information

the experimental platform

And again, in caption

Unless relevant why present in the text all the terms already in the figure…

“The experimental platform mainly includes the”

Section 6

Revise it to “6. Results and Discussion” which includes analysis…

Check Scientific English coherence…

“Therefore, the oil bubble can be regarded as the tracer particles”

I would revise the content expressed in terms of a) and b) information, also including superscript us…

Not really enlightening

Figure 10. Two Consecutive Frames of Preprocessed Images: (a) nth; (b) (n+1)th.

[Also in Figure 12 caption… and leave spaces before units…]

Why separate in paragraphs. It is the same Figure…

Figure 11a shows

Be always clear in every part of the text…

The experimental results are shown in Figure 12.

Use the unit ONCE at the end…

“5.9%, 5.9% and 4.9%,”

Conclusions:

As usually in an abstract:

Brief contextualization then brief methodology

Main findings

Practical implications

Are expected to be found

Please avoid the “listing”…

Instead present relevant, integrated, assertive content

Are these supplementary material to be added to the text?!

Supplementary Materials: The following are available online at www.mdpi.com/xxx/s1, 382 File:English-editing-certificate.

Never seen this before…

Revise…

Funding: Please add: This research

Terrible English… not understandable…

Acknowledgments: The authorship want to Professor Li Yingwei of Yanshan University , because he give good 389 advices on how to program the software of PIV.

References:  none from 2019, please update

The above comments mainly relate formal corrections which will significantly contribute to enhance the text, also, modifying the structure by merging sections.

Author Response

(The authors gave the same response as above.)

Reviewer 4 Report

Please see the review report.

Author Response

(The authors gave the same response as above.)

Round 2

Reviewer 1 Report

Firstly, the authors should produce a document addressing the comments raised by all the reviewers. I could only see the comments raised by myself.

Please add the following corrections:

Line 31 – oil fields, not oil field

Lines 45-71 – Not sure you followed the correct referencing style in the text.

Line 76 – A few scholars (e.g. indicate some of them)

Lind 82 – The above research means which research. Make it clear.

Lind 92 – not clear

Lind 118 – what is ICP technique?

Line 171 – We substituted

Line 200 – not good

Line 240 – from theory? What is the theory?

Line 248 – from theory? What is the theory?

Lind 424 – Li Professor means?

Line 424 – “on how to write program” should be “on computational/mathematical program used in this work”. Include the appropriate program.

Author Response

Dear Ms. April Zhu and Reviewers,

   Thank you for your letter and for the reviewers’ comments concerning our manuscript. Those comments are very helpful for revising and improving our paper.

We have made correction which we hope meet with approval. The changes are highlighted in yellow in my revised manuscript.

Special thanks to you for your good comments.

We hope that these revisions are satisfactory and that the revised version will be acceptable for publication in the journal sensors. Thank you very much for all your help and looking forward to hearing from you soon.
Wish you all the best!

Sincerely yours,
Changfeng Fu

Reviewer 3 Report

Highlight changes in yellow in a next revision, please. No track changes.

Consider comments in the entire text.

Abstract: All abbreviations need to be defined at first-use, both on abstract and in the text, again: PIV

Correct typos: “. Oil” it starts like that… line 30

Or “94.6%. prompting

Check the entire text…

See that English needs additional corrections… plural… “in many 30 oil field,

Only the last name is used in direct references, correct them all…

Scarano, F. [33]

Mentioned before… [Revise all: Last names and then reference number…]

Several cases available…

And correct spacing too: “Xu ,L.et al. [36] improved

When a figure relates published references, add at the end of the caption: modified/adapted from authors names and reference number… as usual.

Not like that: “Figure 2. Schematic diagram of the PIV system [44-45]

and remove if published like presented

Mentioned before…

[Point 10: Reference information regarding figures must be added at the end of the caption, not in the text…

 “a high-speed camera and a computer [32-33].”

See that already published figures add no knowledge and they must be removed if not adapted/modified and then state so, indicating authors names and reference number]

Revise spacing all over the document:

Figure9” or “horizontal;(2)” or “DN20mm;” and many other cases…

Please, and again, leave space before units…

Again, if Figure 9 is in fact two… different captions must be in the same caption…

Only as example of things to be corrected…

Be consistent

 Be coherent

Again, remove all “We” and alike expressions in the manuscript…

we changed the PIV algorithm

Use FIND tool…

Summary and conclusions, is it not the same…?

6. Summary and Conclusions

See the need to improve the text considering previous revision…:

To overcome the problem of low measurement accuracy of the PIV algorithm generated from 397 blank and error vectors in the analysis of flow characteristics of the oil–water two-phase flow with 398 high water cut and low flow velocity in a small horizontal pipe, we changed the PIV algorithm in two 399 regards, a two-dimensional sub-pixel fitting and a velocity vector interpolation, and produced the 400 following conclusions concerning our 2D-KPIV algorithm:

And then no real clear connection..

1)It can correct

And then poor Englisg again

(3)It provide

And no proper spacing…

Avoid the “lists”… please

Although the authors considered the comments of the reviewer, there is a significant number of alterations that still need to be made. English needs additional proofreading.

You may see by the English used at the end, that the text needs further English correction, despite mentioned English editing service…

 “Acknowledgments: We thank Li Professor Yingwei (Yanshan University) for his advice on how to write 424 program.

Besides English editing itself, which really difficult the perception of corrections to be made by the authors, who are not fluent in English and thus no not easily see the need to change…, I believe the text needs some further work to be made relevant. It has to do with many aspects pointed out in previous revision and the ned to better connect content… The last section is an example of that.

Author Response

(The authors gave the same response as above.)

Reviewer 4 Report

The revised manuscript can be published in Sensors.

Author Response

Dear Ms. April Zhu and Reviewers,

   Thank you for your help on our manuscript.

Wish you all the best!

Sincerely yours,
Changfeng Fu

Round 3

Reviewer 3 Report

Highlight changes in yellow in a next revision, please. No track changes.

Consider comments in the entire text.

The text still needs additional proofreading.

Spacing still needs revision in the entire text…

Only as example:

DN20 mm”; “DN20” cannot be a value…

0.8 g/cm3;(6)

And many more…

Figures as Figure 8 have low definition…

There is still one “we” in “Acknowledgments:”. Please use: The authors”

Conclusions section: authors could start by a very brief contextualization, thus leaving clear the importance of this study.

In a scientific text no not use unclear expressions such as “are a lot

Confirm lack of coherence, use this knowledge to anlyse the entire text…

two improvements of PIV algorithm

And then “The improved algorithm

There are many “that” missing in the text…: “This indicates the 2D-KPIV

This expression is nor adequate… “pipe is realized.

Unless the entire text is scientifically and also proofread in terms of English the qualitity of language will not really change…

See that this is not… correct, as many other sentences

Depending on interpretation well logging data of each layer,

Or “the oil layers which do not produce oil is plugged.”~

“the amount of surface (…) are”?!

And “are realized” realized?!

And to end like this…

Moreover, this work can be used in Micro-PIV 425 technique and 3D-PIV technique.

The text just needs complete further intervention in English… to achieve overall coherence.

We can clearly see that the conclusions section is written in a “cyclic” language, goes back to “improvements” again and again.

Language needs to be worked to be clear and assertive… for the text to be concise and relevant.

These are just examples; authors do need to consider the text as a whole…

Author Response

Dear Reviewer,

   Thank you for the comments concerning our manuscript. Those comments are very helpful for revising and improving our paper.

We have made correction which we hope meet with approval. The changes are highlighted in yellow in my revised manuscript.

Special thanks to you for your good comments.

We hope that these revisions are satisfactory and that the revised version will be acceptable for publication in the journal sensors. Thank you very much for all your help and looking

forward to hearing from you soon.
Wish you all the best!

Sincerely yours,
Changfeng Fu
